# A comparative study of randomized response techniques using separate and combined metrics of efficiency and privacy

**Muhammad Azeem** [1]*, **Javid Shabbir**[2,3], **Najma Salahuddin**[4], **Sundus Hussain**[4], **Musarrat Ijaz**[5]

**1** Department of Statistics, University of Malakand, Khyber Pakhtunkhwa, Pakistan, **2** Department of Statistics, University of Wah, Wah Cantt, Pakistan, **3** Department of Statistics, Quaid-i-Azam University, Islamabad, Pakistan, **4** Department of Statistics, Shaheed Benazir Bhutto Women University, Peshawar, Pakistan, **5** Department of Statistics, Rawalpindi Women University, Rawalpindi, Pakistan

* azeemstats@uom.edu.pk

**Data Availability Statement:** All relevant data are available within the manuscript.

## Abstract

In social surveys, the randomized response technique can be considered a popular method for collecting reliable information on sensitive variables. Over the past few decades, it has been a common practice that survey researchers develop new randomized response techniques and show their improvement over previous models. In majority of the available research studies, the authors tend to report only those findings which are favorable to their proposed models. They often tend to hide the situations where their proposed randomized response models perform worse than the already available models. This approach results in biased comparisons between models which may influence the decision of practitioners about the choice of a randomized response technique for real-life problems. We conduct a neutral comparative study of four available quantitative randomized response techniques using separate and combined metrics of respondents' privacy level and model's efficiency. Our findings show that, depending on the particular situation at hand, some models may be better than the other models for a particular choice of values of parameters and constants. However, they become less efficient when a different set of parameter values are considered. The mathematical conditions for efficiency of different models have also been obtained.

## 1. Introduction

In sample surveys, researchers almost always need to cope with high rates of refusals as well as false response respondents when collecting the information related to sensitive traits. Some examples of such sensitive characteristics include cheating behavior in an examination, illegally earned income, tax payable, and abortions, etc. Attempting to devise a strategy to obtain reliable data from the respondents in surveys on sensitive issues, Warner [1] suggested the randomized response procedure. Initially, this procedure was developed to cope with situations in which the researchers have to collect information on qualitative characteristics. Warner [2] modified the originally developed qualitative variable—based procedure to the case of quantitative variables through the introduction of an additive noise. Eichhorn and Hayre [3] proposed an improved variant of the quantitative randomized response strategy by utilizing a multiplicative noise in place of additive noise.

**Funding:** The author(s) received no specific funding for this work.

**Competing interests:** The authors have declared that no competing interests exist.

Gupta et al. [4] presented a randomization technique in which respondents are given an option to provide their true response or use a randomization procedure to report a random response. If any of the respondents chooses to report a scrambled answer, he/she has to apply additive scrambling to answer the sensitive question. An improved multiplicative variant of the Gupta et al. [4] scrambling method was introduced in the research study of Bar-Lev et al. [5]. Later on, Gjestvang and Singh [6] presented an improved optional method which utilized an additive random variable. Diana and Perri [7] presented an improved randomized response strategy by using additive as well as multiplicative random noise. Likewise, Al-Sobhi et al. [8] suggested an additive-subtractive scrambling method for quantitative data.

Gupta et al. [9] proposed a quantification metric which is used for quantification of the privacy level and efficiency into a unified quantity. Later, Narjis and Shabbir [10] proposed a modification of the original Gjestvang and Singh [6] procedure and proved its improvement in terms of efficiency as compared to the randomization technique of Gjestvang and Singh [6]. Khalil et al. [11] carried out a research study to analyze the impact of observational error on estimation of the mean of finite population. Gupta et al. [12] proposed an improved randomization technique and proved its improvement over the Diana and Perri [7] randomization procedure. The comparison was made by taking into account the privacy level as well as model's efficiency.

Singh et al. [13] studied elimination of the influence of non-response using the randomized scrambling technique. Gupta et al. [14] developed an estimator of the population variance using the Diana and Perri [7] randomization strategy. Saleem and Sanaullah [15] suggested estimators of the population mean under randomized response techniques. Zapata et al. [16] suggested an improvement in the Warner's [1] technique. In a recent study, Azeem [17] proposed a weighted measure of efficiency and privacy for assessing the performance of randomized response techniques. Murtaza et al. [18] analyzed randomized response models under the assumption of correlated variables. Further research related to various forms of randomized response techniques can be found in Yan et al. [19], Young et al. [20], Zhang et al. [21], Azeem et al. [22], Azeem [23], and Azeem and Salam [24], among others.

## 2. Selected randomized response models for comparison

Suppose our population under consideration contains a total of $N$ elements and we draw a simple random sample containing $n$ elements from our population. We denote the quantitative sensitive variable of interest by $Y$, and we also consider an additive random variable, say $S$. We also assume that $E(Y_i) = \mu_Y$, $E(S) = \theta$, $V(Y_i) = \sigma_Y^2$, $V(S) = \sigma_S^2$, where $\sigma_Y^2$ and $\sigma_S^2$ denote, respectively, the population variance of $Y$ and $S$; and let $\mu_Y$ and $\theta$ be the notations for the population mean of the variable of interest $Y$ and random variable $S$, respectively. Further, let $T$ be the notation for another scrambling variable, for which we assume that $E(T) = 1$, along with $Var(T) = \sigma_T^2$. Finally, we assume that all three variables under consideration are independent of each other.

### 2.1 The Gjestvang and Singh [6] quantitative model

Using the Gjestvang and Singh [6] optional scrambling model, the observed responses may be expressed as:

$$Z_i = \begin{cases} Y_i + \alpha S, & \text{with probability } p = \dfrac{\beta}{\alpha + \beta} \\[2ex] Y_i - \beta S, & \text{with probability } 1 - p = \dfrac{\alpha}{\alpha + \beta}, \end{cases} \tag{1}$$

where $\alpha$ and $\beta$ denote the constants and are determined by the interviewer.

An estimator of $\mu_Y$ using the Gjestvang and Singh [6] scrambling procedure may be expressed as:

$$\hat{\mu}_{GS} = \frac{1}{n}\sum_{i=1}^{n} Z_i. \tag{2}$$

The variance of $\hat{\mu}_{GS}$ can be derived as:

$$Var(\hat{\mu}_{GS}) = \frac{1}{n}\left[\alpha\beta(\sigma_S^2 + \theta^2) + \sigma_Y^2\right]. \tag{3}$$

## 2.2 Diana and Perri [7] model

The randomization strategy of Diana and Perri [7] contains both additive and multiplicative scrambling. The observed responses based on this model, are given as:

$$Z = TY + S. \tag{4}$$

Assuming $E(S) = 0$, an unbiased estimator of the sensitive variable using the Diana and Perri [7] model can be expressed as:

$$\hat{\mu}_{DP} = \frac{1}{n}\sum_{i=1}^{n} Z_i. \tag{5}$$

The variance of $\hat{\mu}_{DP}$ can be derived as:

$$Var(\hat{\mu}_{DP}) = \frac{1}{n}\left[\sigma_T^2(\sigma_Y^2 + \mu_Y^2) + \sigma_Y^2 + \sigma_S^2\right]. \tag{6}$$

## 2.3 Narjis and Shabbir [10] optional scrambling model

Using the Narjis and Shabbir [10] optional scrambling model, the observed responses can be expressed as:

$$Z_i = \begin{cases} Y_i - \beta S, & \text{with probability } p_1 = \dfrac{\alpha}{\alpha + \beta + \gamma} \\[2mm] Y_i + \alpha S, & \text{with probability } p_2 = \dfrac{\beta}{\alpha + \beta + \gamma} \\[2mm] Y_i, & \text{with probability } p_3 = \dfrac{\gamma}{\alpha + \beta + \gamma}, \end{cases} \tag{7}$$

where $\gamma$ is a constant and its value is chosen by the interviewer before the survey is conducted.

An estimator of $\mu_Y$ using the Narjis and Shabbir [10] optional technique may be written as:

$$\hat{\mu}_{NS} = \frac{1}{n}\sum_{i=1}^{n} Z_i, \tag{8}$$

where $Z_i$ has been provided in Eq (4).

The variance of $\hat{\mu}_{NS}$ can be expressed as:

$$Var(\hat{\mu}_{NS}) = \frac{1}{n}\left[\frac{\alpha\beta(\alpha + \beta)(\sigma_S^2 + \theta^2)}{\alpha + \beta + \gamma} + \sigma_Y^2\right]. \tag{9}$$

## 2.4 Gupta et al. [12] scrambling technique

Using the Gupta et al. [12] scrambling technique, the observed responses are:

$$Z = \begin{cases} Y & \text{with probability } 1 - W \\ Y + S & \text{with probability } WA \\ TY + S & \text{with probability } W(1 - A), \end{cases} \tag{10}$$

where $W$ indicates the sensitivity level, and $A$ denotes a constant such that $0 < A < 1$. Using the Gupta et al. [12] technique and assuming $E(S) = 0$, an unbiased estimator can be expressed as:

$$\hat{\mu}_G = \frac{1}{n} \sum_{i=1}^{n} Z_i. \tag{11}$$

The variance of $\hat{\mu}_G$ can be derived as:

$$Var(\hat{\mu}_G) = \frac{1}{n} \left[ W(1 - A)\sigma_T^2(\sigma_Y^2 + \mu_Y^2) + \sigma_Y^2 + W\sigma_S^2 \right]. \tag{12}$$

## 3. Re-formulating the variance expressions for comparison

For the purpose of unbiased estimation of population mean, the Diana and Perri [7] and the Gupta et al. [12] models assumed that $E(S) = 0$. On the other hand, the Gjestvang and Singh [6] and the Narjis and Shabbir [10] models assumed that $E(S) = \theta$. In order to make the comparison simple and uniform, we assume that $E(S) = 0$ for all four models. Thus, the sampling variance under the Gjestvang and Singh [6] technique may be re-written as:

$$Var(\hat{\mu}_{GS}) = \frac{1}{n} \left[ \alpha\beta\sigma_S^2 + \sigma_Y^2 \right]. \tag{13}$$

In the same way, the variance of the sample mean using the Narjis and Shabbir [10] optional technique may be re-written as:

$$Var(\hat{\mu}_{NS}) = \frac{1}{n} \left[ \frac{\alpha\beta(\alpha + \beta)}{\alpha + \beta + \gamma}\sigma_S^2 + \sigma_Y^2 \right]. \tag{14}$$

The Narjis and Shabbir [10] and the Gupta et al. [12] models used different notations for probabilities for various types of responses. For purpose of comparison, we attempt to bring the mathematical expressions for variance of the mean under identical notations. Equating the probability of true response for the Narjis and Shabbir [10] and the Gupta et al. [12] models, we get:

$$1 - W = \frac{\gamma}{\alpha + \beta + \gamma}, \tag{15}$$

or

$$W = \frac{\alpha + \beta}{\alpha + \beta + \gamma}. \tag{16}$$

Equating the probability of additive scrambling of the Narjis and Shabbir [10] and the Gupta et al. [12] models, we get:

$$WA = \frac{\beta}{\alpha + \beta + \gamma},$$

or

$$\left(\frac{\alpha + \beta}{\alpha + \beta + \gamma}\right)A = \frac{\beta}{\alpha + \beta + \gamma},$$

or

$$A = \frac{\beta}{\alpha + \beta}. \tag{17}$$

Thus

$$1 - A = \frac{\alpha}{\alpha + \beta}, \tag{18}$$

and

$$W(1 - A) = \frac{\alpha + \beta}{\alpha + \beta + \gamma} \times \frac{\alpha}{\alpha + \beta} = \frac{\alpha}{\alpha + \beta + \gamma}. \tag{19}$$

Using Eq (16) to Eq (19), the variance of the mean using the Gupta et al. [12] technique can be written as:

$$Var(\hat{\mu}_G) = \frac{1}{n}\left[\frac{\alpha + \beta}{\alpha + \beta + \gamma}\sigma_S^2 + \sigma_Y^2 + \frac{\alpha}{\alpha + \beta + \gamma}\sigma_T^2(\sigma_Y^2 + \mu_Y^2)\right]. \tag{20}$$

## 4. Performance evaluation

A measure of privacy level was presented by Yan et al. [13] which is given below:

$$\nabla = E[(Z - Y)\hat{2}]. \tag{21}$$

From Eq (21) one may observe that a larger value of $\nabla$ is preferable as it shows a higher level of respondents' privacy.

The Gupta et al. [9] combined metric of privacy level and efficiency can be expressed as:

$$\delta = \frac{MSE}{\nabla}. \tag{22}$$

From Eq (22), it is clear that smaller values of $\delta$ are desirable.

The measure of privacy using the Gjestvang and Singh [6] method can be written as:

$$\nabla_{GS} = \alpha\beta\sigma_S^2. \tag{23}$$

The combined metric of efficiency and privacy level using the Gjestvang and Singh [6] method may be derived as follows:

$$\delta_{GS} = \frac{Var(\hat{\mu}_{GS})}{\nabla_{GS}} = \frac{1}{n}\left[\frac{\alpha\beta\sigma_S^2 + \sigma_Y^2}{\alpha\beta\sigma_S^2}\right]. \tag{24}$$

The measure of privacy level using the Diana and Perri [7] model is given by:

$$\nabla_{DP} = E[TY + S - Y]^2 = \sigma_T^2(\sigma_Y^2 + \mu_Y^2) + \sigma_S^2. \tag{25}$$

The combined metric of privacy and efficiency using the Diana and Perri [7] model can be expressed as:

$$\delta_{DP} = \frac{Var(\hat{\mu}_{DP})}{\nabla_{DP}} = \frac{1}{n}\left[\frac{\sigma_T^2(\sigma_Y^2 + \mu_Y^2) + \sigma_S^2 + \sigma_Y^2}{\sigma_T^2(\sigma_Y^2 + \mu_Y^2) + \sigma_S^2}\right]. \tag{26}$$

Using the Narjis and Shabbir [10] optional model, the metric of privacy-level may be derived as:

$$\nabla_{NS} = \frac{\alpha\beta(\alpha+\beta)}{\alpha+\beta+\gamma}\sigma_S^2. \tag{27}$$

The combined metric of efficiency and privacy level using the Narjis and Shabbir [10] technique can be derived as:

$$\delta_{NS} = \frac{Var(\hat{\mu}_{NS})}{\nabla_{NS}} = \frac{1}{n}\left[\frac{\frac{\alpha\beta(\alpha+\beta)}{\alpha+\beta+\gamma}\sigma_S^2 + \sigma_Y^2}{\frac{\alpha\beta(\alpha+\beta)}{\alpha+\beta+\gamma}\sigma_S^2}\right]. \tag{28}$$

Finally, the metric of privacy level using the Gupta et al. [12] optional technique can be written as:

$$\nabla_G = \frac{\alpha}{\alpha+\beta}\left[\sigma_T^2(\sigma_Y^2 + \mu_Y^2)\right] + \sigma_S^2. \tag{29}$$

The combined metric of the privacy level and model efficiency using the Gupta et al. [12] technique can be obtained as:

$$\delta_G = \frac{Var(\hat{\mu}_G)}{\nabla_G} = \frac{1}{n}\left[\frac{\frac{\alpha+\beta}{\alpha+\beta+\gamma}\sigma_S^2 + \sigma_Y^2 + \frac{\alpha}{\alpha+\beta+\gamma}\sigma_T^2(\sigma_Y^2 + \mu_Y^2)}{\frac{\alpha}{\alpha+\beta}\sigma_T^2(\sigma_Y^2 + \mu_Y^2) + \sigma_S^2}\right]. \tag{30}$$

## 5. Conditions for efficiency

Here we present the mathematical expression for conditions of efficiencies of various models.

### 5.1 Gupta et al. [12] quantitative model vs. Narjis and Shabbir [10] scrambling model

The Gupta et al. [12] scrambling technique is more precise than the Narjis and Shabbir [10] technique, if:

$$Var(\hat{\mu}_G) < Var(\hat{\mu}_{NS}),$$

or

$$\frac{1}{n}\left[\frac{\alpha+\beta}{\alpha+\beta+\gamma}\sigma_S^2 + \sigma_Y^2 + \frac{\alpha}{\alpha+\beta+\gamma}\sigma_T^2(\sigma_Y^2 + \mu_Y^2)\right] < \frac{1}{n}\left[\frac{\alpha\beta(\alpha+\beta)}{\alpha+\beta+\gamma}\sigma_S^2 + \sigma_Y^2\right],$$

or

$$\alpha\sigma_T^2(\sigma_Y^2 + \mu_Y^2) < (\alpha\beta - 1)(\alpha+\beta)\sigma_S^2,$$

or

$$\frac{\alpha}{(\alpha\beta-1)(\alpha+\beta)} < \frac{\sigma_S^2}{\sigma_T^2(\sigma_Y^2 + \mu_Y^2)}. \tag{31}$$

### 5.2 Gupta et al. [12] quantitative model vs. Gjestvang and Singh [6] quantitative model

The Gupta et al. [12] technique is more precise than the Gjestvang and Singh [6] model, if:

$$Var(\hat{\mu}_G) < Var(\hat{\mu}_{GS}),$$

or

$$\frac{1}{n}\left[\frac{\alpha+\beta}{\alpha+\beta+\gamma}\sigma_S^2 + \sigma_Y^2 + \frac{\alpha}{\alpha+\beta+\gamma}\sigma_T^2(\sigma_Y^2+\mu_Y^2)\right] < \frac{1}{n}\left[\alpha\beta\sigma_S^2 + \sigma_Y^2\right],$$

or

$$\left(\frac{\alpha+\beta}{\alpha+\beta+\gamma} - \alpha\beta\right)\sigma_S^2 + \frac{\alpha}{\alpha+\beta+\gamma}\sigma_T^2(\sigma_Y^2+\mu_Y^2) < 0,$$

or

$$\frac{\alpha}{\alpha\beta(\alpha+\beta+\gamma) - (\alpha+\beta)} < \frac{\sigma_S^2}{\sigma_T^2(\sigma_Y^2+\mu_Y^2)}. \tag{32}$$

### 5.3 Gupta et al. [12] model vs. Diana and Perri [7] model

The Gupta et al. [12] technique is more precise than the Diana and Perri [7] model, if

$$Var(\hat{\mu}_G) < Var(\hat{\mu}_{DP}),$$

or

$$\frac{1}{n}\left[\frac{\alpha+\beta}{\alpha+\beta+\gamma}\sigma_S^2 + \sigma_Y^2 + \frac{\alpha}{\alpha+\beta+\gamma}\sigma_T^2(\sigma_Y^2+\mu_Y^2)\right] < \frac{1}{n}\left[\sigma_T^2(\sigma_Y^2+\mu_Y^2) + \sigma_S^2 + \sigma_Y^2\right],$$

or

$$(\beta+\gamma)\sigma_T^2(\sigma_Y^2+\mu_Y^2) + \gamma\sigma_S^2 > 0. \tag{33}$$

Condition (33) is always true.

### 5.4 Narjis and Shabbir [10] scrambling model vs. Diana and Perri [7] model

The Narjis and Shabbir [10] scrambling model will be more precise than the Diana and Perri [7] model, if

$$Var(\hat{\mu}_{NS}) < Var(\hat{\mu}_{DP}),$$

or

$$\frac{1}{n}\left[\frac{\alpha\beta(\alpha+\beta)}{\alpha+\beta+\gamma}\sigma_S^2 + \sigma_Y^2\right] < \frac{1}{n}\left[\sigma_T^2(\sigma_Y^2+\mu_Y^2) + \sigma_S^2 + \sigma_Y^2\right],$$

or

$$\frac{\alpha\beta(\alpha+\beta) - (\alpha+\beta+\gamma)}{\alpha+\beta+\gamma} < \frac{\sigma_T^2(\sigma_Y^2+\mu_Y^2)}{\sigma_S^2}. \tag{34}$$

## 5.5 Narjis and Shabbir [10] model vs. Gjestvang and Singh [6] scrambling model

The scrambling model suggested by Narjis and Shabbir [10] will be more precise than the Gjestvang and Singh [6] model, if

$$Var(\hat{\mu}_{NS}) < Var(\hat{\mu}_{GS}),$$

or

$$\frac{1}{n}\left[\frac{\alpha\beta(\alpha+\beta)}{\alpha+\beta+\gamma}\sigma_S^2 + \sigma_Y^2\right] < \frac{1}{n}\left[\alpha\beta\sigma_S^2 + \sigma_Y^2\right],$$

or

$$\frac{\alpha+\beta}{\alpha+\beta+\gamma} < 1. \tag{35}$$

Condition (35) is always true, since $\gamma > 0$.

## 5.6 Diana and Perri [7] model vs. Gjestvang and Singh [6] model

The Diana and Perri [7] model is more precise than the Gjestvang and Singh [6] model, if

$$Var(\hat{\mu}_{DP}) < Var(\hat{\mu}_{GS}),$$

or

$$\frac{1}{n}\left[\sigma_T^2(\sigma_Y^2 + \mu_Y^2) + \sigma_S^2 + \sigma_Y^2\right] < \frac{1}{n}\left[\alpha\beta\sigma_S^2 + \sigma_Y^2\right],$$

or

$$\alpha\beta - 1 > \frac{\sigma_T^2(\sigma_Y^2 + \mu_Y^2)}{\sigma_S^2},$$

or

$$\alpha\beta > \frac{\sigma_S^2 + \sigma_T^2(\sigma_Y^2 + \mu_Y^2)}{\sigma_S^2}. \tag{36}$$

## 6. A real-world survey

We applied the four selected techniques to a practical student survey, selecting 40 undergraduate students from the students registered in the Department of Mathematics of the University of Malakand, Pakistan. We were interested in estimating the average grade point average (GPA) of the students. Each selected participant was given a deck of 100 cards as well as a calculator. Each card displayed a random number for each of the two scrambling variables $T$ and $S$, generated from a normal distribution. For the scrambling variable $S$, we chose the mean of the normal distribution zero with variance 0.5. Likewise, for variable $T$, the mean of the distribution was 1 with variance 0.5.

The survey procedure using a randomized response technique has been presented in Fig 1.

The values of the constants $\alpha$, $\beta$, and $\gamma$ were chosen based on some prior knowledge about the population under study. If prior information is not available, a pilot survey may be conducted to obtain an estimate of the constants. For this survey, we decided to choose $\alpha = 3$, $\beta = 3$ and $\gamma = 4$, so that $1 - W = \frac{\gamma}{\alpha+\beta+\gamma} = 0.4$, $WA = 0.6 \times 0.5 = 0.3$, and

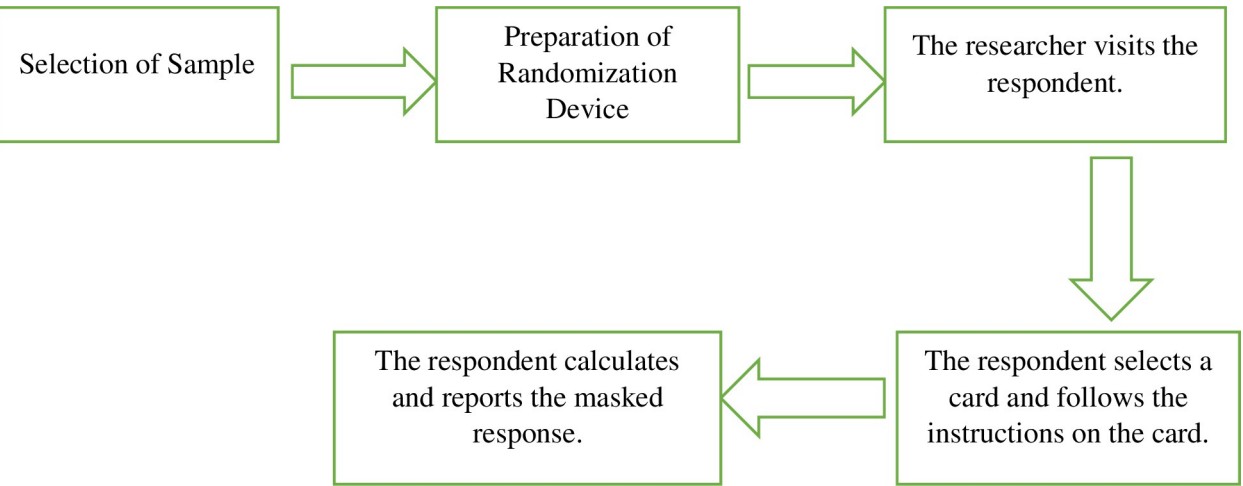

**Fig 1. Randomized response survey procedure.**

$W(1 - A) = 0.6 \times 0.5 = 0.3$. Using these choices of constants, the observed response function under the Gjestvang and Singh [6] model may be expressed as follows:

$$Z = \begin{cases} Y + 3S, & \text{with probability } 0.5, \\ Y - 3S, & \text{with probability } 0.5. \end{cases} \tag{37}$$

The observed response under the Diana and Perri [7] model may be expressed as follows:

$$Z = TY + S. \tag{38}$$

The observed response using the Narjis and Shabbir [10] model may be expressed as follows:

$$Z = \begin{cases} Y - 3S, & \text{with probability } 0.3, \\ TY + 3S, & \text{with probability } 0.3, \\ Y, & \text{with probability } 0.4. \end{cases} \tag{39}$$

The observed response using the Gupta et al. [12] scrambling method may be expressed as follows:

$$Z = \begin{cases} Y, & \text{with probability } 0.4, \\ Y + S, & \text{with probability } 0.3, \\ TY + S, & \text{with probability } 0.3. \end{cases} \tag{40}$$

Using Eq (37), each card displayed one of the two types of instructions for the Gjestvang and Singh [6] model:

50 out of 100 cards had the instruction: "Add 3 times the value of S with your GPA and report the number you get."

The remaining 50 cards had the instruction: "Subtract 3 times the value of S from your GPA and report the number you get."

In the same manner, the survey was repeated four times with different instructions on each card for each model. Each student was instructed to choose one card at random and read the instructions on the selected card to report the response. To ensure privacy protection, the participants were also asked neither to disclose the card they selected, nor their true GPA. The

**Table 1. Observed responses using the Gjestvang and Singh [6] model.**

| -1.813 | 3.083 | -0.987 | 0.539 | 6.782 | 2.319 | 4.731 | -1.964 | 3.660 | 2.623 |
|--------|-------|--------|-------|-------|-------|-------|--------|-------|-------|
| 2.856 | 1.418 | 5.889 | 3.685 | 2.618 | 1.949 | 5.109 | 3.931 | -2.023 | 4.632 |
| 4.979 | -2.639 | 3.931 | 2.816 | 0.949 | 2.865 | 1.525 | 2.430 | 1.987 | 3.894 |
| 3.881 | 3.101 | 2.193 | -1.980 | 4.953 | 1.736 | 2.654 | 5.761 | 0.896 | -1.120 |

**Table 2. Observed responses using the Diana and Perri [7] model.**

| 3.514 | 3.643 | 2.674 | 1.563 | 1.967 | 4.310 | 3.751 | 0.961 | 3.536 | 1.429 |
|-------|-------|-------|-------|-------|-------|-------|-------|-------|-------|
| 1.812 | 2.498 | 4.879 | 3.187 | 3.528 | 0.842 | 1.609 | 1.987 | 2.603 | 5.712 |
| 4.519 | 5.659 | 2.032 | 2.866 | 2.347 | 1.753 | 3.564 | 4.519 | 1.424 | 1.829 |
| 2.784 | 0.807 | 3.196 | 4.975 | 2.423 | 3.736 | 1.477 | 2.753 | 3.195 | 2.015 |

**Table 3. Observed responses using the Narjis and Shabbir [10] model.**

| 0.617 | 3.073 | 2.714 | -1.633 | 4.986 | 2.316 | 5.221 | 3.961 | -1.230 | 3.824 |
|-------|-------|-------|--------|-------|-------|-------|-------|--------|-------|
| -1.862 | 2.828 | -0.860 | 4.685 | 2.918 | 4.940 | 1.349 | -1.323 | 3.902 | -1.782 |
| 3.551 | -2.630 | 4.738 | 2.876 | 3.048 | 3.965 | 2.460 | 3.479 | 4.829 | -1.195 |
| 1.873 | 3.901 | 3.684 | 1.963 | 4.741 | 1.435 | -0.678 | 4.961 | 2.697 | 3.917 |

**Table 4. Observed responses using the Gupta et al. [12] model.**

| 2.524 | 0.244 | 3.770 | 1.960 | 4.983 | 0.518 | 3.751 | -0.267 | 3.733 | 1.728 |
|-------|-------|-------|-------|-------|-------|-------|--------|-------|-------|
| 0.916 | 5.458 | 2.489 | 3.154 | 3.528 | 1.817 | 1.429 | 3.885 | 2.921 | 0.632 |
| 4.318 | 3.656 | 0.042 | 1.226 | 2.347 | 2.253 | 0.864 | 5.511 | 1.754 | 3.939 |
| 1.874 | 1.917 | 3.398 | 4.273 | 2.423 | 1.726 | 4.457 | 0.854 | 3.297 | 2.111 |

respondents were only required to report the scrambled / masked value. The responses recorded from the participants under different models are presented in Tables 1–4.

It may be observed that the responses recorded under the Diana and Perri [7] model are all positive numbers, ranging from 0.807 to 5.712. The feasible range of students GPA is 0 to 4, however, the scrambling process resulted in a few out-of-range responses. The Gupta et al. [12] model provided only one negative response, with the responses ranging from -0.267 to 5.458. It is also observed that the Gjestvang and Singh [6] model and the Narjis and Shabbir [10] model provided several negative and out-of-range responses. The sample means of the data in Tables 1–4 are 2.35, 2.85, 2.31, and 2.54, respectively.

## 7. Numerical illustration

Table 5 displays the sampling variance under different models for various choice of values of $\alpha, \beta$, and $\gamma$. Tables 6 and 7 present $\nabla$ and $\delta$ values, respectively, using various randomized response models.

## 8. Discussion and conclusion

The present study analyzed a detailed comparison among four available quantitative randomized response techniques: (i) Gjestvang and Singh [6] model, (ii) Diana and Perri [7] model, (iii) Narjis and Shabbir [10] scrambling technique, and (iv) Gupta et al. [12] optional model.

**Table 5. Variance of the sample mean using various models when $\mu_Y = 10$, $\sigma_Y = 2$, and $n = 50$.**

| $\alpha$ | $\beta$ | $\gamma$ | $\sigma_T = 0.4$ | | | | $\sigma_T = 1$ | | | |
|---|---|---|---|---|---|---|---|---|---|---|
| | | | GS | DP | NS | G | GS | DP | NS | G |
| 1 | 3 | 5 | 0.215 | 2.413 | 0.140 | 0.137 | 0.215 | 4.16 | 0.140 | 0.331 |
| | | 8 | 0.215 | 2.413 | 0.125 | 0.123 | 0.215 | 4.16 | 0.125 | 0.268 |
| | 5 | 8 | 0.305 | 2.413 | 0.176 | 0.123 | 0.305 | 4.16 | 0.176 | 0.248 |
| | | 12 | 0.305 | 2.413 | 0.155 | 0.113 | 0.305 | 4.16 | 0.155 | 0.211 |
| 4 | 6 | 10 | 1.160 | 2.413 | 0.620 | 0.169 | 1.160 | 4.16 | 0.620 | 0.483 |
| | | 15 | 1.160 | 2.413 | 0.512 | 0.151 | 1.160 | 4.16 | 0.512 | 0.408 |
| | 10 | 20 | 1.880 | 2.413 | 0.821 | 0.138 | 1.880 | 4.16 | 0.821 | 0.315 |
| | | 30 | 1.880 | 2.413 | 0.653 | 0.125 | 1.880 | 4.16 | 0.653 | 0.267 |
| 5 | 3 | 1 | 0.755 | 2.413 | 0.680 | 0.305 | 0.755 | 4.16 | 0.680 | 1.276 |
| | | 1.5 | 0.755 | 2.413 | 0.648 | 0.293 | 0.755 | 4.16 | 0.648 | 1.213 |
| | 5 | 2 | 1.205 | 2.413 | 1.018 | 0.256 | 1.205 | 4.16 | 1.018 | 0.984 |
| | | 4 | 1.205 | 2.413 | 0.884 | 0.231 | 1.205 | 4.16 | 0.884 | 0.855 |
| 8 | 10 | 15 | 3.680 | 2.413 | 2.044 | 0.185 | 3.680 | 4.16 | 2.044 | 0.609 |
| | | 25 | 3.680 | 2.413 | 1.587 | 0.161 | 3.680 | 4.16 | 1.587 | 0.486 |
| | 15 | 20 | 5.480 | 2.413 | 2.968 | 0.166 | 5.480 | 4.16 | 2.968 | 0.491 |
| | | 30 | 5.480 | 2.413 | 2.423 | 0.150 | 5.480 | 4.16 | 2.423 | 0.413 |
| 10 | 4 | 2 | 1.880 | 2.413 | 1.655 | 0.327 | 1.880 | 4.16 | 1.655 | 1.419 |
| | | 3 | 1.880 | 2.413 | 1.562 | 0.313 | 1.880 | 4.16 | 1.562 | 1.341 |
| | 8 | 3 | 3.680 | 2.413 | 3.166 | 0.277 | 3.680 | 4.16 | 3.166 | 1.109 |
| | | 6 | 3.680 | 2.413 | 2.780 | 0.252 | 3.680 | 4.16 | 2.780 | 0.980 |
| 20 | 10 | 4 | 9.080 | 2.413 | 8.021 | 0.315 | 9.080 | 4.16 | 8.021 | 1.343 |
| | | 8 | 9.080 | 2.413 | 7.185 | 0.291 | 9.080 | 4.16 | 7.185 | 1.210 |
| | 15 | 8 | 13.580 | 2.413 | 11.068 | 0.271 | 13.580 | 4.16 | 11.068 | 1.084 |
| | | 12 | 13.580 | 2.413 | 10.133 | 0.255 | 13.580 | 4.16 | 10.133 | 0.999 |

GS: $Var(\hat{\mu}_{GS})$, DP: $Var(\hat{\mu}_{DP})$, NS: $Var(\hat{\mu}_{NS})$, G: $Var(\hat{\mu}_G)$

The mathematical conditions for efficiency comparison of various models were obtained. We found that some of the efficiency conditions are always true, whereas other conditions are not always true. Table 6 shows that when $\sigma_T = 0.4$, Gupta et al. [12] technique appears to be the most efficient among all four randomized response techniques. However, for $\sigma_T = 1$, the Narjis and Shabbir [10] randomization technique is more precise than the Gupta et al. [12] model for a variety of choices of $\alpha$, $\beta$, and $\gamma$. It is also clear that since the Gjestvang and Singh [6] and the Narjis and Shabbir [10] models do not use multiplicative scrambling variable, $T$, so the variance of the mean under these models does not change when $\sigma_T$ changes. Moreover, the variance of the mean under Diana and Perri [7] model does not change when the values of $\alpha$, $\beta$, and $\gamma$ change. It is also observed that the Diana and Perri [7] scrambling model is the worst among all four models in terms of efficiency.

The quality of randomized response models cannot be solely judged from model-efficiency. The respondents' privacy protection is also an important aspect for judging the usefulness of randomized response techniques. The level of privacy may be quantified by the values of $\nabla$ where a larger value indicates better privacy level. Table 1 shows the $\nabla$ values for various values of $\alpha$, $\beta$, and $\gamma$. It also indicates that for $\sigma_T = 0.4$, the Gjestvang and Singh [6] optional model is the best among all four models when privacy protection of the respondents is taken into account. The performance of Diana and Perri [7] model is also observed to be better than the

**Table 6. Results of $\nabla$ values under various models when $\mu_Y = 10$, $\sigma_Y = 2$, and $n = 50$.**

| $\alpha$ | $\beta$ | $\gamma$ | $\sigma_T = 0.4$ | | | | $\sigma_T = 1$ | | | |
|---|---|---|---|---|---|---|---|---|---|---|
| | | | $\nabla_{GS}$ | $\nabla_{DP}$ | $\nabla_{NS}$ | $\nabla_{G}$ | $\nabla_{GS}$ | $\nabla_{DP}$ | $\nabla_{NS}$ | $\nabla_{G}$ |
| 1 | 3 | 5 | 6.75 | 18.89 | 3 | 6.41 | 6.75 | 106.25 | 3 | 28.25 |
| | | 8 | 6.75 | 18.89 | 2.25 | 6.41 | 6.75 | 106.25 | 2.25 | 28.25 |
| | 5 | 8 | 11.25 | 18.89 | 4.82 | 5.02 | 11.25 | 106.25 | 4.82 | 19.58 |
| | | 12 | 11.25 | 18.89 | 3.75 | 5.02 | 11.25 | 106.25 | 3.75 | 19.58 |
| 4 | 6 | 10 | 54 | 18.89 | 27 | 8.91 | 54 | 106.25 | 27 | 36.92 |
| | | 15 | 54 | 18.89 | 21.6 | 8.91 | 54 | 106.25 | 21.6 | 36.92 |
| | 10 | 20 | 90 | 18.89 | 37.06 | 7 | 90 | 106.25 | 37.06 | 24.14 |
| | | 30 | 90 | 18.89 | 28.64 | 7 | 90 | 106.25 | 28.64 | 24.14 |
| 5 | 3 | 1 | 33.75 | 18.89 | 30 | 12.65 | 33.75 | 106.25 | 30 | 67.25 |
| | | 1.5 | 33.75 | 18.89 | 28.42 | 12.65 | 33.75 | 106.25 | 28.42 | 67.25 |
| | 5 | 2 | 56.25 | 18.89 | 46.88 | 10.57 | 56.25 | 106.25 | 46.88 | 54.25 |
| | | 4 | 56.25 | 18.89 | 40.18 | 10.57 | 56.25 | 106.25 | 40.18 | 54.25 |
| 8 | 10 | 15 | 180 | 18.89 | 98.18 | 9.65 | 180 | 106.25 | 98.18 | 48.47 |
| | | 25 | 180 | 18.89 | 75.35 | 9.65 | 180 | 106.25 | 75.35 | 48.47 |
| | 15 | 20 | 270 | 18.89 | 144.42 | 8.04 | 270 | 106.25 | 144.42 | 38.42 |
| | | 30 | 270 | 18.89 | 117.17 | 8.04 | 270 | 106.25 | 117.17 | 38.42 |
| 10 | 4 | 2 | 90 | 18.89 | 78.75 | 14.14 | 90 | 106.25 | 78.75 | 76.54 |
| | | 3 | 90 | 18.89 | 74.12 | 14.14 | 90 | 106.25 | 74.12 | 76.54 |
| | 8 | 3 | 180 | 18.89 | 154.29 | 11.49 | 180 | 106.25 | 154.29 | 60.03 |
| | | 6 | 180 | 18.89 | 135 | 11.49 | 180 | 106.25 | 135 | 60.03 |
| 20 | 10 | 4 | 450 | 18.89 | 397.06 | 13.34 | 450 | 106.25 | 397.06 | 71.58 |
| | | 8 | 450 | 18.89 | 355.26 | 13.34 | 450 | 106.25 | 355.26 | 71.58 |
| | 15 | 8 | 675 | 18.89 | 549.42 | 11.76 | 675 | 106.25 | 549.42 | 61.68 |
| | | 12 | 675 | 18.89 | 502.66 | 11.76 | 675 | 106.25 | 502.66 | 61.68 |

Gupta et al. [12] model. However, when $\sigma_T = 1$, the Diana and Perri [7] model becomes the best among all models for most of the cases of values of $\alpha$, $\beta$, and $\gamma$.

Finally, comparing the *overall usefulness* of the four quantitative models using $\delta$ values, the results are shown in Table 7. It is clearly observed in Table 3 that the Diana and Perri [7] model performs best with respect to $\delta$ values when $\sigma_T = 0.4$. However, for $\sigma_T = 1$, the Gupta et al. [12] scrambling method reduces the values of $\delta$ to a minimum level, which makes it the most useful model of all four models in terms of overall quality.

We conclude that a randomized response model which can perform best in one situation may perform the worst in another situation. So, in practical problems, the researchers should be aware of the situation at hand when deciding to choose a particular randomized response model for collecting data from respondents. The researchers may choose a randomized response technique when respondent-privacy is more important than efficiency. On the other hand, if model-efficiency is more important to the researcher than privacy protection, a different randomized response model may be more useful.

The present study compared four available randomized response models. We found that, depending on the choice of parameters, one model can perform better than another model, and vice versa. The current study is limited to quantitative models, however, in many practical problems, the variable under consideration may be of qualitative nature. Therefore, it may be interesting if a neutral comparative analysis of qualitative models is carried out. Moreover, the current study is limited to the case of no correlation among variables. In practice, some degree

Table 7. $\delta$ values under various models when $\mu_Y = 10$, $\sigma_Y = 2$, and $n = 50$.

| $\alpha$ | $\beta$ | $\gamma$ | $\sigma_T = 0.4$ | | | | $\sigma_T = 1$ | | | |
|---|---|---|---|---|---|---|---|---|---|---|
| | | | $\delta_{GS}$ | $\delta_{DP}$ | $\delta_{NS}$ | $\delta_G$ | $\delta_{GS}$ | $\delta_{DP}$ | $\delta_{NS}$ | $\delta_G$ |
| 1 | 3 | 5 | 0.03185 | 0.012773 | 0.04667 | 0.02137 | 0.03185 | 0.03915 | 0.04667 | 0.01172 |
| | | 8 | 0.03185 | 0.012773 | 0.05556 | 0.01915 | 0.03185 | 0.03915 | 0.05556 | 0.00950 |
| | 5 | 8 | 0.02711 | 0.012773 | 0.03659 | 0.02450 | 0.02711 | 0.03915 | 0.03659 | 0.01266 |
| | | 12 | 0.02711 | 0.012773 | 0.04133 | 0.02259 | 0.02711 | 0.03915 | 0.04133 | 0.01075 |
| 4 | 6 | 10 | 0.02148 | 0.012773 | 0.02296 | 0.01898 | 0.02148 | 0.03915 | 0.02296 | 0.01308 |
| | | 15 | 0.02148 | 0.012773 | 0.02370 | 0.01698 | 0.02148 | 0.03915 | 0.02370 | 0.01106 |
| | 10 | 20 | 0.02089 | 0.012773 | 0.02216 | 0.01966 | 0.02089 | 0.03915 | 0.02216 | 0.01306 |
| | | 30 | 0.02089 | 0.012773 | 0.02279 | 0.01779 | 0.02089 | 0.03915 | 0.02279 | 0.01107 |
| 5 | 3 | 1 | 0.02237 | 0.012773 | 0.02267 | 0.02410 | 0.02237 | 0.03915 | 0.02267 | 0.01897 |
| | | 1.5 | 0.02237 | 0.012773 | 0.02281 | 0.02317 | 0.02237 | 0.03915 | 0.02281 | 0.01803 |
| | 5 | 2 | 0.02142 | 0.012773 | 0.02171 | 0.02424 | 0.02142 | 0.03915 | 0.02171 | 0.01814 |
| | | 4 | 0.02142 | 0.012773 | 0.02199 | 0.02185 | 0.02142 | 0.03915 | 0.02199 | 0.01576 |
| 8 | 10 | 15 | 0.02044 | 0.012773 | 0.02081 | 0.01920 | 0.02044 | 0.03915 | 0.02081 | 0.01256 |
| | | 25 | 0.02044 | 0.012773 | 0.02106 | 0.01667 | 0.02044 | 0.03915 | 0.02106 | 0.01002 |
| | 15 | 20 | 0.02030 | 0.012773 | 0.02055 | 0.02065 | 0.02030 | 0.03915 | 0.02055 | 0.01278 |
| | | 30 | 0.02030 | 0.012773 | 0.02068 | 0.01863 | 0.02030 | 0.03915 | 0.02068 | 0.01076 |
| 10 | 4 | 2 | 0.02089 | 0.012773 | 0.02102 | 0.02316 | 0.02089 | 0.03915 | 0.02102 | 0.01855 |
| | | 3 | 0.02089 | 0.012773 | 0.02108 | 0.02213 | 0.02089 | 0.03915 | 0.02108 | 0.01752 |
| | 8 | 3 | 0.02044 | 0.012773 | 0.02052 | 0.02410 | 0.02044 | 0.03915 | 0.02052 | 0.01848 |
| | | 6 | 0.02044 | 0.012773 | 0.02059 | 0.02196 | 0.02044 | 0.03915 | 0.02059 | 0.01633 |
| 20 | 10 | 4 | 0.02018 | 0.012773 | 0.02020 | 0.02364 | 0.02018 | 0.03915 | 0.02020 | 0.01876 |
| | | 8 | 0.02018 | 0.012773 | 0.02023 | 0.02178 | 0.02018 | 0.03915 | 0.02023 | 0.01691 |
| | 15 | 8 | 0.02012 | 0.012773 | 0.02015 | 0.02308 | 0.02012 | 0.03915 | 0.02015 | 0.01758 |
| | | 12 | 0.02012 | 0.012773 | 0.02016 | 0.02170 | 0.02012 | 0.03915 | 0.02016 | 0.01619 |

of correlation may exist among variables which may affect the findings of the comparison. We therefore recommend future researchers to perform a comparative assessment of randomized response models assuming correlated variables as it may give further interesting results.

## Supporting information

**S1 Data.**
(XLSX)

## Author Contributions

**Conceptualization:** Muhammad Azeem.

**Data curation:** Javid Shabbir, Musarrat Ijaz.

**Formal analysis:** Muhammad Azeem, Javid Shabbir, Najma Salahuddin.

**Investigation:** Najma Salahuddin.

**Methodology:** Javid Shabbir, Sundus Hussain.

**Software:** Muhammad Azeem, Najma Salahuddin.

**Supervision:** Muhammad Azeem.

**Validation:** Javid Shabbir, Sundus Hussain, Musarrat Ijaz.

**Writing – original draft:** Muhammad Azeem.

**Writing – review & editing:** Javid Shabbir, Najma Salahuddin, Sundus Hussain, Musarrat Ijaz.

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
