## [Decision Letter · Decision Letter 0]

21 Aug 2023

PONE-D-23-20411A comparative study of randomized response techniques using separate and combined metrics of efficiency and privacyPLOS ONE

Dear Dr. Azeem,

Thank you for submitting your manuscript to PLOS ONE. After careful consideration, we feel that it has merit but does not fully meet PLOS ONE’s publication criteria as it currently stands. Therefore, we invite you to submit a revised version of the manuscript that addresses the points raised during the review process.

We look forward to receiving your revised manuscript.

Kind regards,

Viacheslav Kovtun, Dr.Sc., Ph.D.

Academic Editor

PLOS ONE

Journal Requirements:

Reviewers' comments:

Reviewer's Responses to Questions

**Comments to the Author**

1. Is the manuscript technically sound, and do the data support the conclusions?

Reviewer #1: Partly

Reviewer #2: Yes

2. Has the statistical analysis been performed appropriately and rigorously? 

Reviewer #1: Yes

Reviewer #2: N/A

3. Have the authors made all data underlying the findings in their manuscript fully available?

Reviewer #1: No

Reviewer #2: Yes

4. Is the manuscript presented in an intelligible fashion and written in standard English?

Reviewer #1: Yes

Reviewer #2: Yes

5. Review Comments to the Author

Reviewer #1: In this manuscript, the authors conduct a neutral analysis of four available randomized response methods, making their findings versatile and beneficial to a wide audience. Despite the good use of the English language and a good structure of the paper, many concepts need to be focused on improving paper quality:

- In the introductory section, the contributions, limitations and novelty of this work are not clear or little specified. I suggest you try to include more bibliographical references for dealing with this topic to make a proper comparison and under light your contributions to the research;

- Explain better how the study looks at only four quantitative randomized response techniques, which might not cover the broad spectrum of available methods.

- The emphasis on the efficiency of models based on "a particular choice of values of parameters and constants" means that these findings might not be universally applicable. This might require practitioners to undergo an additional step of parameter fine-tuning before settling on a model. How the authors can overcome this issue?

- While the study does a comparative analysis, it might be enhanced by the application or demonstration of these techniques on real-world survey data.

- In the conclusion section, I suggest supporting the achieved results by offering a more critical/discursive view of future research.

Reviewer #2: There are some points to be clarified:

1. Eq (5) and (11)

In order to have these estimators to be unbiased, E[S] should be 0. However, the description of E[S] = 0 is delayed after Eq (12).

2. Eq (21)

The right hand should be written as E[(Z-Y)^2].

3. To better present the paper, the author should consider giving a pictorial description of the survey procedure. Especially, it is unclear who gives the scrambling value, and whether the respondents can submit the masked value only or not.

6. PLOS authors have the option to publish the peer review history of their article (what does this mean?). If published, this will include your full peer review and any attached files.

Reviewer #1: **Yes: **Giovanni Cicceri

Reviewer #2: **Yes: **Hiroshi Toyoizumi

---

## [Author Response · Author response to Decision Letter 0]

29 Aug 2023

REVISION REPORT ON PONE-D-23-20411

Dear editor, I am submitting the revised version of our manuscript PONE-D-23-20411. I am thankful to you for inviting to submit a revised version of the paper. I am also thankful to the honorable referees for valuable suggestions to improve the quality of the paper. Here, I am providing a point-by-point response to the suggestions / questions asked by both reviewers. I hope you and the reviewers will be satisfied from the revision.

Reviewer 1:

1. In the introductory section, the contributions, limitations and novelty of this work are not clear or little specified. I suggest you try to include more bibliographical references for dealing with this topic to make a proper comparison and under light your contributions to the research. 

Response: The paper has been revised to clearly specify the contribution and novelty in the abstract. In the revised paper, the abstract contains these sentences:

Research Background: In social surveys, the randomized response technique can be considered a popular method for collecting reliable information on sensitive variables. Over the past few decades, it has been a common practice that survey researchers develop new randomized response techniques and show their improvement over previous models.

Research Gap: In majority of the available research studies, the authors tend to report only those findings which are favorable to their proposed models. They often tend to hide the situations where their proposed randomized response models perform worse than the already available models. This approach results in biased comparisons between models which may influence the decision of practitioners about the choice of a randomized response technique for real-life problems.

Our Work: We conduct a neutral comparative study of four available quantitative randomized response techniques using separate and combined metrics of respondents’ privacy level and model’s efficiency.

Our Findings: Our findings show that, depending on the particular situation at hand, some models may be better than the other models for a particular choice of values of parameters and constants. However, they become less efficient when a different set of parameter values are considered. The mathematical conditions for efficiency of different models have also been obtained.

I hope the revised abstract clarifies the research gap, contribution, and novelty of the work. 

The limitations of the study along with future research suggestions have been provided in the last paragraph in the conclusion section. A few more citations of some recent papers have been added to the introduction section (changes highlighted).

2. Explain better how the study looks at only four quantitative randomized response techniques, which might not cover the broad spectrum of available methods.

Response: We have selected four available models from the literature for our comparative analysis. Our selected models include two recent models of Narjis and Shabbir (2021) and Gupta et al. (2022) and two older models of Gjestvang and Singh (2009) and Diana and Perri (2011). There are a variety of randomized response models available in literature, ranging from qualitative to quantitative to mixed models. Since the number / type of scrambling variable(s) and the underlying assumptions vary from model to model, so it may be difficult to cover all of the available models in a single study. For example, the Murtaza et al. (2020) model assumes correlation among variables, whereas the Gupta et al. (2022) model assumes uncorrelated variables. However, we have provided a detailed comparison of the four selected models using separate and combined measures of efficiency and privacy. Further suggestions from reviewers in this regard will be welcomed.

In the conclusion section, we have given suggestions for future research. In future, we have recommended to the researchers to investigate a neutral comparison of qualitative models. Moreover, the case of correlated variables may also be considered for comparison of models.

3. The emphasis on the efficiency of models based on "a particular choice of values of parameters and constants" means that these findings might not be universally applicable. This might require practitioners to undergo an additional step of parameter fine-tuning before settling on a model. How the authors can overcome this issue?

Response: We want to convey the same message to survey practitioners, who may erroneously think that a newer model is always better than an older model. No model can always be better for all possible values of parameters. In the revised manuscript, the wording / grammar of the abstract has been revised to clarify the purpose of the study. Since our study just compares available models and it doesn’t suggest any new model, so we don’t care which model is better for which choice of parameters.

The purpose of this study is to make a comparison of four available randomized response models. Our comparison is neutral, that is, we do not favor one model over the other. In past few decades, a lot of randomized response models have been introduced by researchers. In almost every study, the authors tend to show only the favorable results by hiding the situations where their suggested model may be less efficient than the previous models. This approach by the researchers leads to biased comparison which may misguide the practitioners when choosing a model for data collection. The practitioners tend to blindly follow the authors’ recommendations that the new model is better than the older models. However, our study finds that this may not always be true. For example, we have found that the recently developed Gupta et al. (2022) model may be less efficient than the older model of Narjis and Shabbir (2009) for some choices of parameters, and more efficient for other values of parameters. Many of the recently developed models which the authors have shown more efficient than the previous models, may be less efficient than the older models when the values of the parameters are changed. It all depends on fine-tuning of parameters which the survey practitioners often ignore when choosing a particular model for a practical problem.

Moreover, model efficiency is not the only criterion of assessing model quality. The measure of respondents’ privacy is also an important consideration when evaluating randomized response models. Our study suggests to the practitioners to consider all aspects of model quality when choosing a randomized response model out of many available models for practical problems.

4. While the study does a comparative analysis, it might be enhanced by the application or demonstration of these techniques on real-world survey data.

Response: A real-world data collection example using the four selected models has been added in Section 6. The addition of the real-world survey example in the revised manuscript explains data collection using these four models. The authors who originally developed these four models had not provided any real-world application of their models.

5. In the conclusion section, I suggest supporting the achieved results by offering a more critical/discursive view of future research.

Response: A discursive view of the future research has been added at the end of conclusion section. The following paragraph has been added to the conclusion section:

The present study compared four available randomized response models. We found that, depending on the choice of parameters, one model may be better than the other model, and vice versa. The current study is limited to quantitative models, however, in many practical problems, the variable under consideration may be of qualitative nature. Therefore, it may be interesting if a neutral comparative analysis of qualitative models is carried out. Moreover, the current study is limited to the case of no correlation among variables. In practice, some degree of correlation may exist among variables which may affect the findings of the comparison. We therefore recommend future researchers to conduct a comparative study of randomized response models assuming correlated variables as it may give further interesting results.

Reviewer 2:

1. Eq (5) and (11). In order to have these estimators to be unbiased, E[S] should be 0. However, the description of E[S] = 0 is delayed after Eq (12).

Response: ‘Assuming ’ has been added to the sentences before Eq. (5) and Eq. (11).

2. Eq (21). The right hand should be written as E[(Z-Y)^2].

Response: Corrected.

3. To better present the paper, the author should consider giving a pictorial description of the survey procedure. Especially, it is unclear who gives the scrambling value, and whether the respondents can submit the masked value only or not.

Response: A figure describing the survey procedure and a data collection example has been added (see Section 6 in the revised manuscript). The real-world survey example added to the revised manuscript explains the survey procedure. The respondents are required to only report the masked value.

The revised manuscript with track changes has been uploaded to the submission system.

---

## [Decision Letter · Decision Letter 1]

17 Oct 2023

A comparative study of randomized response techniques using separate and combined metrics of efficiency and privacy

PONE-D-23-20411R1

Dear Dr. Azeem,

We’re pleased to inform you that your manuscript has been judged scientifically suitable for publication and will be formally accepted for publication once it meets all outstanding technical requirements.

Kind regards,

Viacheslav Kovtun, Dr.Sc., Ph.D.

Academic Editor

PLOS ONE

Additional Editor Comments (optional):

Reviewers' comments:

Reviewer's Responses to Questions

**Comments to the Author**

1. If the authors have adequately addressed your comments raised in a previous round of review and you feel that this manuscript is now acceptable for publication, you may indicate that here to bypass the “Comments to the Author” section, enter your conflict of interest statement in the “Confidential to Editor” section, and submit your "Accept" recommendation.

Reviewer #1: All comments have been addressed

Reviewer #2: All comments have been addressed

2. Is the manuscript technically sound, and do the data support the conclusions?

Reviewer #1: Yes

Reviewer #2: Yes

3. Has the statistical analysis been performed appropriately and rigorously? 

Reviewer #1: Yes

Reviewer #2: N/A

4. Have the authors made all data underlying the findings in their manuscript fully available?

Reviewer #1: (No Response)

Reviewer #2: Yes

5. Is the manuscript presented in an intelligible fashion and written in standard English?

Reviewer #1: Yes

Reviewer #2: Yes

6. Review Comments to the Author

Reviewer #1: The paper has been revised to improve the clarity and organization of the presentation of the findings. Both the experiments and conclusion sections have been refined to elevate the paper's quality.

Reviewer #2: (No Response)

7. PLOS authors have the option to publish the peer review history of their article (what does this mean?). If published, this will include your full peer review and any attached files.

Reviewer #1: **Yes: **Giovanni Cicceri

Reviewer #2: **Yes: **Hiroshi Toyoizumi

---

## [Editor Report · Acceptance letter]

19 Oct 2023

PONE-D-23-20411R1 

A comparative study of randomized response techniques using separate and combined metrics of efficiency and privacy 

Dear Dr. Azeem:

I'm pleased to inform you that your manuscript has been deemed suitable for publication in PLOS ONE. Congratulations! Your manuscript is now with our production department. 

Kind regards, 

on behalf of

Prof. Viacheslav Kovtun 

Academic Editor

PLOS ONE